# Characteristics and Drivers of Reference Evapotranspiration in Hilly Regions in Southern China

**Youcun Liu** [1,*] **, Yan Liu** [1,*] **, Ming Chen** [1] **, David Labat** [2] **, Yongtao Li** [1,3] **, Xiaohui Bian** [4] **and Qianqian Ding** [4]

[1]   School of Resources and Environmental Engineering, Jiangxi University of Science and Technology, No.156, Kejia Avenue, Ganzhou 341000, China; just168@163.com (M.C.); yongtao@scau.edu.cn (Y.L.)
[2]   Géosciences Environnement Toulouse (GET), Université de Toulouse (CNRS, IRD, OMP), 14 Avenue Edouard Belin, 31400 Toulouse, France; david.labat@get.omp.eu
[3]   School of Natural Resources and Environment, South China Agricultural University, No.483, Wushan Avenue, Guangzhou 510642, China
[4]   School of Architectural and Surveying &Mapping Engineering, Jiangxi University of Science and Technology, Ganzhou 341000, China; xiaohuibian@hotmail.com (X.B.); qianqding@hotmail.com (Q.D.)
*   Correspondence: liuyc@lzb.ac.cn (Yo.L.); ly2017lhye@126.com (Ya.L.); Tel.: +86-155-7006-3891 (Yo.L.)

**Abstract:** This paper has adopted related meteorological data collected by 69 meteorological stations between 1951 and 2013 to analyze changes and drivers of reference evapotranspiration ($ET_0$) in the hilly regions located in southern China. Results show that: (1) $ET_0$ in southern China's hilly regions reaches its maximum in summer and its minimum in winter, and that the annual $ET_0$ shows an increasing trend. $ET_0$ happened abrupt change due to the impact of abrupt meteorological variables changes, and the significant year of mutation were 1953, 1964 and 2008. Most abrupt changes of $ET_0$ in meteorological stations occurred in the 1950s and 1960s. (2) The low value of $ET_0$ was mainly captured in high-altitude areas. Spatially, the $ET_0$ in the east was higher than that in the west. With the exception of a handful of stations, the trend coefficients of $ET_0$ were all positive, exhibiting a gradual rise. Changes in $ET_0$ in the east were much more sensitive than that in the west. Since $ET_0$ was affected by the cyclical changes in relative humidity, short-period oscillations were observed in all these changes. (3) In general, the ET0 was negatively correlated with relative humidity, and positively correlated with temperature and sunshine percentage. $ET_0$ is most sensitive to changes in average temperature, with a sensitivity coefficient of 1.136. $ET_0$ showed positive sensitivity to average temperature and sunshine hours, which were notable in the northeastern, and uniform in the spatial. $ET_0$ showed negatively sensitivity to relative humidity, and the absolute value of sensitivity coefficient in the northwestern is smaller. The highest contribution to $ET_0$ is the average temperature (6.873%), and the total contribution of the four meteorological variables to the change of $ET_0$ is 7.842%. The contribution of average temperature, relative humidity, and sunshine hours to $ET_0$ is higher in the northern and eastern, northern, northern and eastern areas, respectively. Climate indexes (Western Pacific Index (WP), Southern Oscillation Index (SOI), Tropical Northern Atlantic Index (TNA), and El Niño-Southern Oscillation (ENSO)) were correlated with the $ET_0$. In addition, the $ET_0$ and altitude, as well as the latitude and longitude were also correlated with each other.

**Keywords:** reference evapotranspiration; sensitivity coefficient; contribution; hilly regions in southern China

## 1. Introduction

As the population grows, the threat of climate change to global food security has become a major challenge in the 21st century [1]. Evapotranspiration (ET) is an important means of surface water consumption, and it is also an important variable of the hydrological cycle in the atmosphere terrestrial system [2]. The changes of evapotranspiration influence various regional water equilibrium components, including the production and life in the region, and agriculture production especially. Reference evapotranspiration ($ET_0$) is a reflection of the energy available to evaporate water under adequate water supply conditions. Meteorological factors, such as wind speed, temperature, solar radiation and precipitation have various changes under the conditions of global warming and climate change in China, and these factors are closely related to changes in reference evapotranspiration [3]. With wind available to transport the water vapor from the ground up into the lower atmosphere, evapotranspiration accounts for the movement of water within a plant, and the subsequent loss of water as vapor through stomata in its leaves. Reference evapotranspiration is said to equal actual evapotranspiration when there is ample water, and it is an important part of the water cycle [4]. Although the actual evapotranspiration is an objective variable for measuring water changes, the lack of long-term and large-scale actual evapotranspiration data makes the study of reference evapotranspiration important for understanding the actual evapotranspiration [5]. As an important indicator to measure the regional water status, reference evapotranspiration not only has an impact on the sustainable development of the social economy, but also has a profound impact on regional agricultural production, especially in terms of food security [6]. Therefore, a scientific basis can be provided for the Equationtion of China's agricultural change by analyzing the dynamic and spatial characteristics of reference evapotranspiration in the assessment area, and exploring its response to global climate change.

Quite a few studies have been conducted on reference evapotranspiration in the world, (especially in selected ecologically fragile areas). The research on reference evapotranspiration is mainly in terms of two aspects. One is the temporal and spatial variation of reference evapotranspiration, and the other is the influence of meteorological elements on the reference evapotranspiration. Some scholars focus on the analysis of spatiotemporal variation of reference evapotranspiration. Wang et al. [7] researched the variety characters reference evapotranspiration in the Zoige wetland, and Xu et al. [8] analyze the spatial distribution and temporal trends in the Changjiang catchment of China. There are still many scholars who are concerned with the analysis of the influencing factors of reference evapotranspiration. While Yang et al. [9] adopted the Thornthwaite and Penman–Monteith methods to conduct sensitivity analysis on global reference evapotranspiration, Roderick et al. [10] concluded that evaporation in the Northern Hemisphere underwent an overall reduction over the past 50 years and discussed the possible causes. De la Casa et al. [11] studied the change of reference evapotranspiration in central Argentina. Results shows that beneficial change in agricultural suitability reported for Central Argentina region was almost exclusively produced by increasing plus precipitation. Croitoru et al. [12] used the monthly meteorological data of 57 sites in Romania to analyze the variation characteristics of reference evapotranspiration under the change of meteorological variables in Romania from 1961 to 2007. In addition, some scholars have also studied the reference evapotranspiration in the Northeastern [13,14], Northwestern [15] and Southwestern areas [16] as well as, the Changjiang Basin [8,17], of China. Huang Huiping [18–20] and other scholars have even pointed out that China's annual average reference evapotranspiration underwent an overall reduction over the past 50 years. In short, due to the harsh climatic conditions and fragile ecological environment in northern China [21], especially the northwestern region, numerous studies on surface water have been successfully conducted in the northwestern region. However, the research is often overlooked in terms on agricultural climate resources and water balance particularly in the Jiangnan hilly areas in the humid climate zone due to the so-called "excellent climatic conditions and good ecological environment." The precipitation is abundant in the Jiangnan hilly area, as the most important grain producing area in China. However, it has seen an upward trend of the reference evapotranspiration and extreme

hydrological events, such as droughts and floods in higher frequency and on a larger scale in recent decades [22,23]. The purpose of the paper is to analyze the spatial and temporal trends of reference evapotranspiration and the driving factors of reference evapotranspiration in the Jiangnan hilly area for more than 60 years. In this regard, hence the study on the reference evapotranspiration in the hilly areas of the south of the Yangtze River can provide a scientific basis for the Equationtion of regional agricultural planning and agricultural development strategies and agricultural response to climate change.

*Overview of Research Areas*

Hilly regions in southern China usually refer to the areas in south of the Yangtze River, north of Nanling, east of Xuefeng Mountain, and west of Mount Wuyi (Figure 1), which is approximately about 370,000 km$^2$ in area. As these areas fall within the typical subtropical monsoon climate zone, they are characterized by hot and rainy summers, as well as mild and dry winters. The annual precipitation is about 1300–1800 mm while annual average temperature is about 16–20 °C. Research areas feature abundant water sources, numerous rivers and lakes, mainly including the Dongting Lake Basin and Poyang Lake Basin. Dongting Lake Plain and Poyang Lake Plain, with a long history in agriculture production, are also significant grain producing areas in southern China.

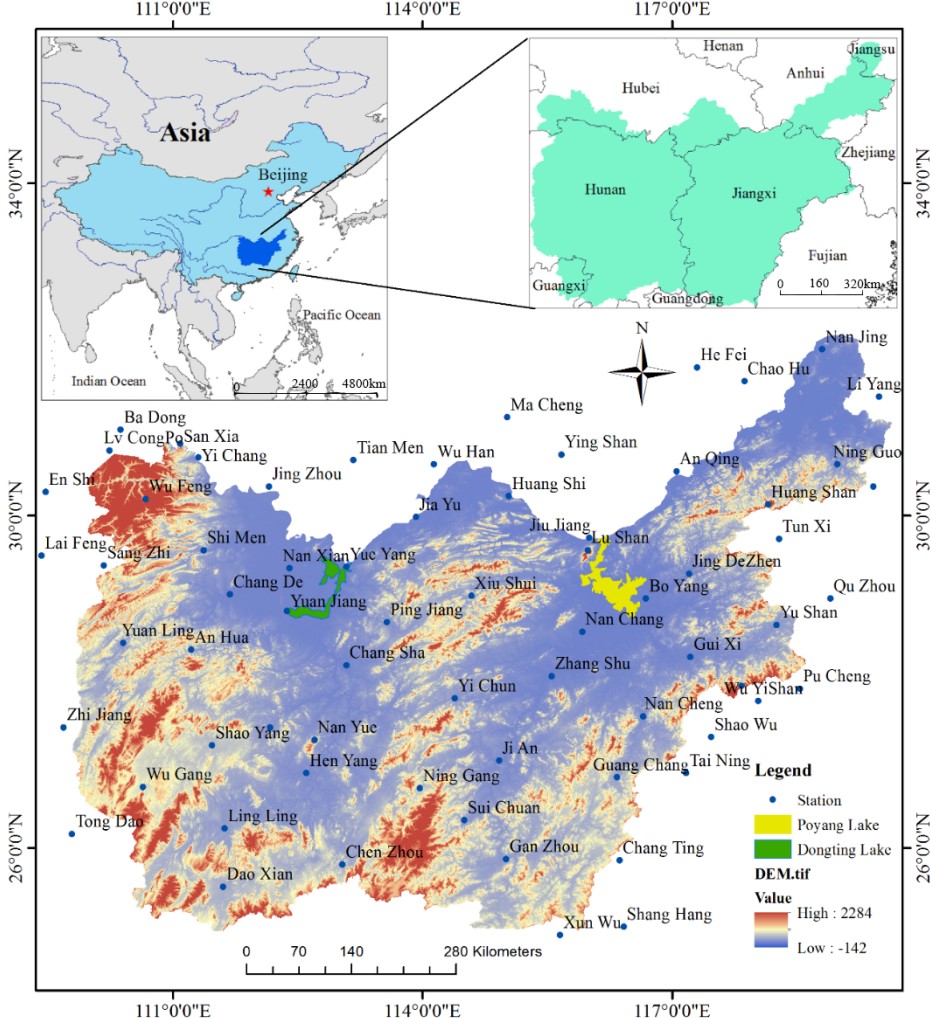

**Figure 1.** Map of Research Areas.

## 2. Data and Methods

### *2.1. Data Source*

This paper has selected various meteorological data collected between 1951 and 2013 by 69 meteorological stations located in the hilly regions in southern China. The data includes different meteorological variables such as temperature, wind speed, relative humidity and sunshine hours, etc. Meteorological data had been provided by the National Meteorological Science Data Sharing Network.

### *2.2. Research Methods*

#### 2.2.1. Reference Evapotranspiration

Several methods/Equations, such as Penman–Monteith [24], Hargreaves and Samani [25], Thornthwaite [26] and Makkink [27], can be applied to calculate reference evapotranspiration. At present, the Penman–Monteith method, recommended by FAO (Food and Agriculture Organization of the United Nations) in 1998, is the most widely adopted one. This paper has also applied the Penman–Monteith method to calculate reference evapotranspiration, hence taking various meteorological elements, including temperature, wind speed, relative humidity and sunshine hours, etc., into account.

$$\text{ET}_0 = \frac{0.408 \times \Delta \times (R_n - G) + \frac{900\gamma \times u_2 \times (e_s - e_a)}{T+273}}{\Delta + \gamma \times (1 + 0.34u_2)} \tag{1}$$

In Equation (1), $\text{ET}_0$ represents reference evapotranspiration (mm/day), $R_n$ the net surface radiation amount (MJm$^{-2}$ day$^{-1}$), G the soil heat flux (MJm$^{-2}$ day$^{-1}$), T the average temperature (°C), $u_2$ the wind speed (ms$^{-1}$) at the height of 2 m, $e_s$ the saturated vapor pressure (KPa), $e_a$ the actual vapor pressure (KPa), $e_s - e_a$ the saturated vapor pressure difference (KPa), $\Delta$ the slope of the vapor pressure curve (KPa) °C$^{-1}$), and $\gamma$ the constant of the hygrometer.

There are many methods for calculating net radiation. The calculation method of net radiation used in this paper is the method of net short-wave radiation minus net long-wave radiation:

$$R_n = R_{ns} - R_{nl} \tag{2}$$

where $R_{ns}$ is short-wave radiation and $R_{nl}$ represent long-wave radiation. Also:

$$R_{ns} = (1-a)\left(a_s + b_s\left(\frac{n}{N}\right)\right)R_a \tag{3}$$

where *a* is surface reflectivity, and the value is 0.23; $R_a$ means solar radiation at the top of the atmosphere (MJm$^{-2}$ day$^{-1}$); *n* is sunshine hours (h); *N* is maximum sunshine hours (h); $a_s$ represents the amount of solar radiation reaching the surface of the Earth in cloudy days (*n* = 0); $b_s$ imply the amount of solar radiation reaching the surface of the earth in a sunny day (*n* = *N*); the recommended values of $a_s$ is 0.25 and $b_s$ is 0.50 in this paper.

$$R_a = \frac{24(60)}{\pi} G_{sc} d_r [\omega_s \sin\varphi \cos\sigma + \cos\varphi \sin\sigma \sin\omega_s] \tag{4}$$

where $G_{SC}$ is solar constant (0.0820 MJm$^{-2}$ min$^{-1}$), $d_r$ represents reciprocal of the relative distance between the sun and the earth, $w_s$ is the solar time angle (rad), $\varphi$ is the geographic latitude and $\delta$ is the solar magnetic declination (rad).

$R_{nl}$ is long-wave radiation, the long-wave radiation is affected by various variables such as water vapor and temperature. According to the Stefan–Boltzmann law, the long-wave radiation calculation Equation used in this paper is:

$$R_{nl} = \sigma\left[\frac{T_{max,k}^4 + T_{min,k}^4}{4}\right]\left(0.34 - 0.14\sqrt{e_a}\right)\left[0.1 + 0.9\left(\frac{n}{N}\right)\right] \tag{5}$$

where σ is the Stefan–Boltzmann constant ($4.90 \times 10^{-9}$ MJK$^{-4}$ m$^{-2}$ day$^{-1}$), $T_{max,k}^4$ is the maximum absolute temperature; $T_{min,k}^4$ is the lowest absolute temperature (K = °C + 273.6), $e_a$ is the actual vapor pressure (KPa).

$$G = C_S(T_I - T_{i-1})\Delta Z \tag{6}$$

where G is soil heat flux (MJ m$^{-2}$ day$^{-1}$), $C_S$ is the soil heat capacity (2.1 MJ m$^{-3}$ °C$^{-1}$), $\Delta Z$ is effective soil depth (0.2 m), $T_I$ means the average air temperature of the *i*th day, and $T_{i-1}$ is the average temperature of the *(i−1)*th day.

### 2.2.2. Accumulative Anomaly

Cumulative anomaly is a method used to visually judge the trend of climate change by curve, and its accumulative anomaly is an index to distinguish the changing tendency of discrete data. It is the accumulation of the difference between a value and the average of a series of values. The accumulative anomaly can be calculated by the Equation (7);

$$LP_i = \sum_1^N \left(R_i - \overline{R}\right) \tag{7}$$

where $LP_i$ is the anomaly cumulative value of the *i*-th year; $R_i$ is the ET$_0$ of the *i*-th year; $\overline{R}$ is the mean value of the series $R_i$, and $N$ is the number of discrete points. Concerning the positive and negative characteristics of the anomaly, when the L*Pi* continues to increase, it indicates that the anomaly continues to be positive during the period, that is, the ET$_0$ is higher than the average; When the L*Pi* decreases to zero, it indicates that the anomaly of the period is constant, that is, the ET$_0$ is kept at an average level. When the L*Pi* continues to decrease, it indicates that the anomaly continues to be negative during the period, that is, the ET$_0$ is lower than the average level. Therefore, the interannual variation phase of ET$_0$ can be determined more intuitively and accurately [28].

### 2.2.3. Climatic Trend Coefficients

The study of climatic change trends is based on climatic trend coefficients, hence changes in these coefficients will indicate any increment/drop in various meteorological elements [14]. The value interval is [−1,1]. The closer $r_{xt}$ is to −1, the more significant the drop is; the closer $r_{xt}$ is to 1, the more significant the increment is.

$$r_{xt} = \frac{\sum_{i=1}^n (x_i - \overline{x})(i - \overline{t})}{\sqrt{\sum_{i=1}^n (x_i - \overline{x})^2 \sum_{i=1}^n \left(i - \overline{t}\right)^2}} \tag{8}$$

In Equation (8), $x_i$ represents the element's value in year *i*, and $\overline{x}$ is the sample average; when $\overline{t} = (n+1)/2$, and $r_{xt}$ is positive (negative), it indicates that the element tends to undergo a linear increment (drop) within *n* years.

### 2.2.4. Sensitivity Coefficient and Contribution

The calculation method of the sensitivity coefficient in this paper is the method proposed by McCucn [29]. It is used to calculate the sensitivity coefficient of reference evapotranspiration to key climatic variables, by using the partial derivative of ET$_0$ for each meteorological factor.

$$S_{Vi} = \lim_{V_i \to 0}\left(\frac{ET_0}{\frac{\Delta V_i}{V_i}}\right) = \frac{\partial ET_0}{\partial V_i} \times \frac{V_i}{ET_0} \tag{9}$$

where $S_{Vi}$ is the sensitivity coefficient; $\Delta ET_0$ is the variations of $ET_0$. $V_i$, $\Delta V_i$ is meteorological variables and their variations respectively. If $S_{Vi} > 0$, it means that the reference evapotranspiration increased as the variable increases. If $S_{Vi} < 0$, it indicates that the reference evapotranspiration decreased as the variable increases. The larger $S_{Vi}$, the greater the effect a given variable has on reference evapotranspiration [17]. Based on literature review, and as used by other workers [30], meteorological parameters (temperature, wind speed, sunshine hours and relative humidity) have been selected for a range of specific variability (±20%) for the analysis.

Multiplying the sensitivity coefficient of a single meteorological factor by the multi-year relative change of the variable can obtain the change of reference evapotranspiration caused by this variable, that is, the contribution of the element to the change of reference evapotranspiration [31]. The specific expression is as follows:

$$Con_{V_i} = S_{V_i} \times RC_{V_i} \tag{10}$$

$$RC_{V_i} = \frac{n \times Trend}{|av|} \times 100\% \tag{11}$$

where, $Con_{Vi}$ is the contribution of meteorological factor $Vi$ to $ET_0$ change, $S_{Vi}$ is the sensitivity coefficient of $V_i$, $RC_{Vi}$ (%) is the relative change of $V_i$ for many years, $a_v$ is the multi-year average of $V_i$, and Trend is the year-to-year rate of change of $V_i$, Trend is calculated by the trend analysis method [32].

Reference evapotranspiration is affected by multiple meteorological variables. In this paper, the contribution of the four variables of average temperature, relative humidity, sunshine hours and wind speed is added to obtain the total contribution to the change of reference evapotranspiration [33]. The Equation is:

$$Con_{ET_0} = Con_T + Con_{RH} + Con_n + Con_U \tag{12}$$

where, $Con_{ET0}$ represents the change of $ET_0$ caused by the interaction of four meteorological variables, also called the estimated change of $ET_0$. $Con_T$, $Con_{RH}$, $Con_n$, and $Con_U$ represent the contribution of average temperature, relative humidity, sunshine hours, and wind speed to changes in $ET_0$ respectively.

### 2.2.5. Wavelet Analysis

Wavelet analysis (also known as multi-resolution analysis) is a commonly used method for analyzing the scale and trend of time series, studying the evolution of different scales (cycles) over time, with multi-resolution analysis and adaptive characteristics to the signal because the periodic transformation characteristics of meteorological elements are complex. It contains periodic changes of various time scales in the same period, and they will exhibit multiple time scale features. Therefore, the use of wavelet analysis to study the evolution of meteorological elements at different scales (cycles) over time has become an important method for studying the long-term changes of meteorological elements [34].

## 3. Result and Analysis

### 3.1. Time Changes in Reference Evapotranspiration

#### 3.1.1. Seasonal Changes and Annual Average Change in Meteorological Variables

Figures 2 and 3 shows the seasonal changes and annual average in meteorological variables from the 69 stations in the hilly regions in southern China over the last 63 years. Average temperature (T) increased with time throughout all seasons, and the most significant increase occurred in spring, and the slope is 0.039. In the past 63 years, relative humidity (RH) showed a fluctuating and declining trend. But, the decline trend of spring is significantly faster than other seasons, the slope is −0.1287. Wind speed (U) in the research areas also showed a fluctuating and declining trend, moreover, significant decline occurred in winter, with a slope of −0.125. Although the change trend of sunshine percentage

(SP) in spring was slowly rising with a slope of 0.0238, the sunshine percentage in other seasons and annual average were decreased with time.

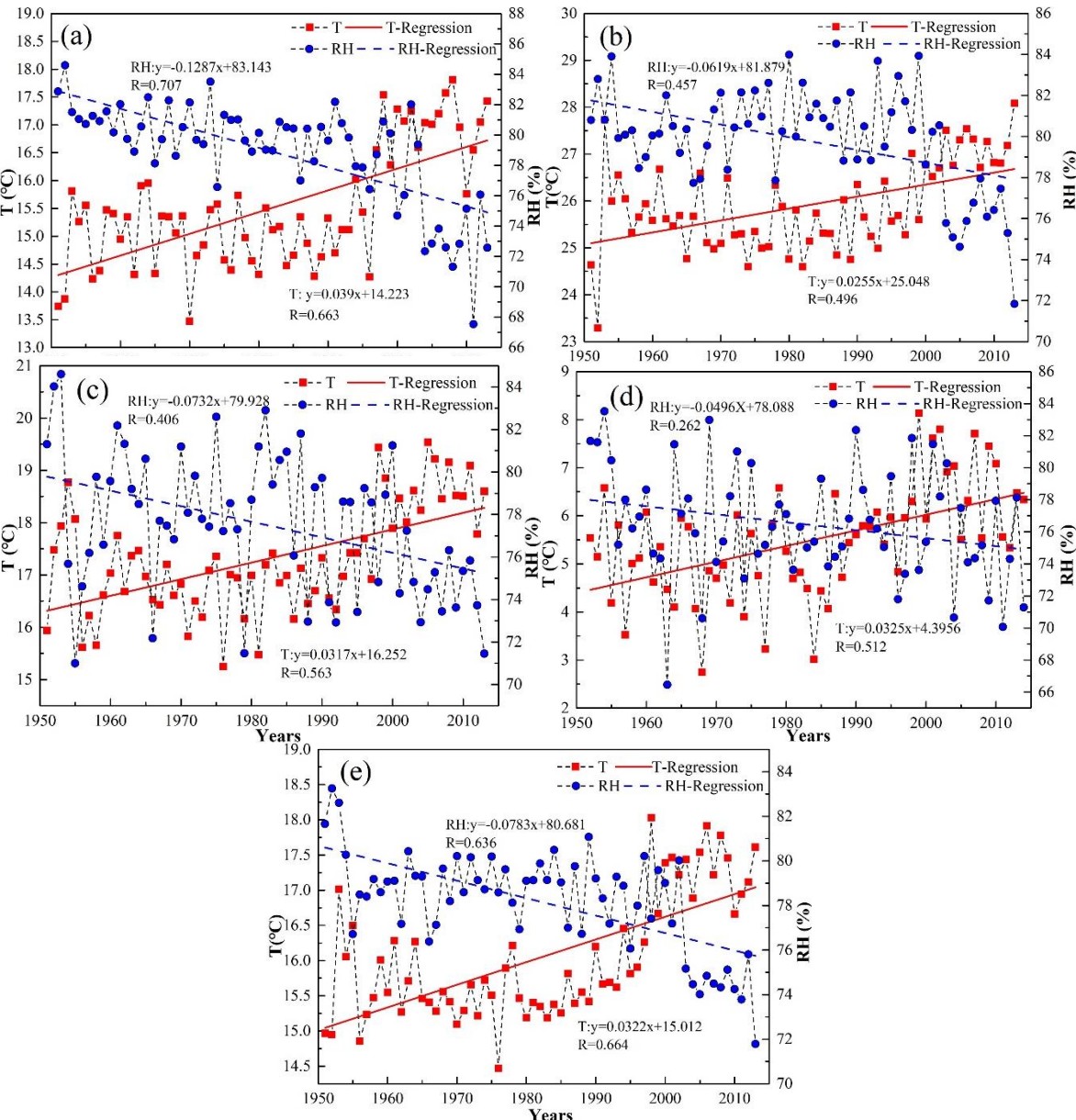

**Figure 2.** Seasonal ((**a**) spring, (**b**) summer, (**c**) autumn, (**d**) winter, (**e**) annual) and Annual Average of T (average temperature), RH (relative humidity).

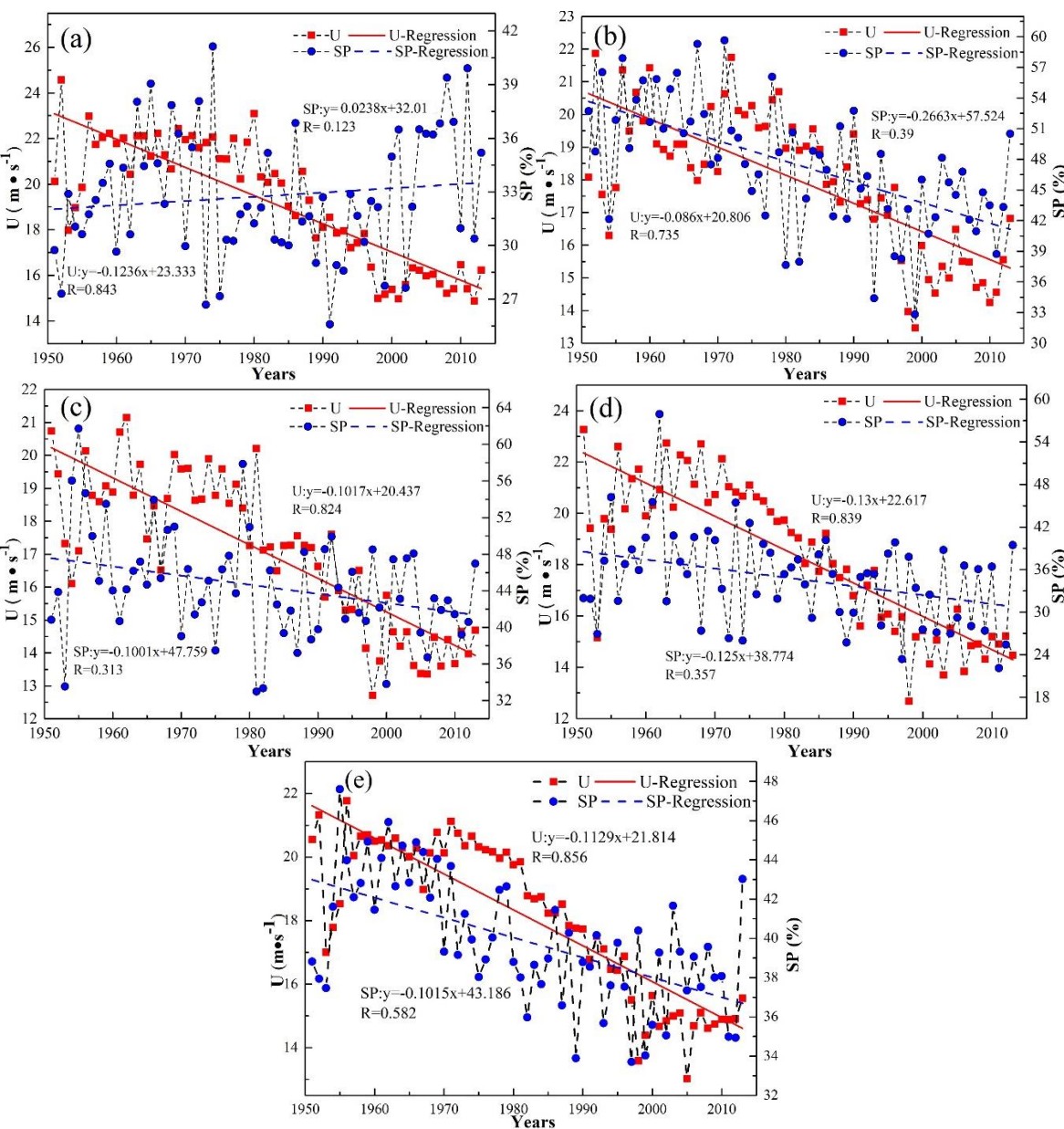

**Figure 3.** Seasonal ((**a**) spring, (**b**) summer, (**c**) autumn, (**d**) winter, (**e**) annual) and Annual Average of U (wind speed), SP (sunshine percentage).

### 3.1.2. Seasonal Changes and Annual Average Change in Reference Evapotranspiration

Figure 4 shows the seasonal changes and annual average change in reference evapotranspiration in the hilly regions in southern China over the last 63 years. A large gap between the reference evapotranspiration in summer and winter (the peak in summer and the lowest point in winter) can be observed due to the large differences in sunlight and temperature in summer and winter. From the linear change trend, reference evapotranspiration increased with time throughout all seasons. However, the most significant change occurred in spring with the highest correlation coefficient (Figure 4a).

Between 1951 and 2013, reference evapotranspiration showed a fluctuating and rising trend, passing the significance test of 0.01 (Figure 4b). The average reference evapotranspiration for several years was 1284 mm, with its highest value in 2013 (indicating a surface evapotranspiration peak) and lowest value in 1952 (indicating the lowest point in surface evapotranspiration). Although average reference evapotranspiration had seen increments and reductions over 5 years, it revealed an overall

fluctuating and rising trend, and it increased significantly in 2001, mainly due to the increase in solar radiation after 2001, resulting in significant fluctuations in temperature and evapotranspiration [35]. Furthermore, changes in cumulative anomalies also reflect the changing trend of the research object. The reference evapotranspiration cumulative anomaly refers to the accumulation of the difference between reference evapotranspiration values and the multi-year average reference evapotranspiration during the calculation period. Between 1951 and 2013, the reference evapotranspiration cumulative anomaly curve went through three main stages: gradual fluctuating rise (before 1966), gradual fluctuating decline (1966 to 2002), rapid rise (after 2002). According to these three stages, $ET_0$ was segmentally fitted, and partition fitting also indicated the changing trend of reference evapotranspiration featuring a rise-slow decline-sharp rise pattern.

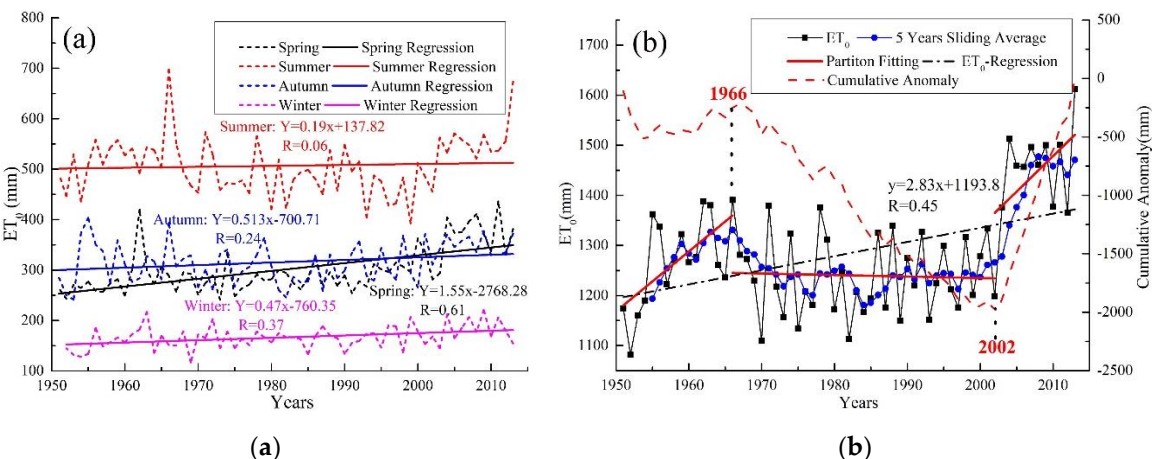

**Figure 4.** Seasonal (**a**) and Annual Average of Reference Evapotranspiration (**b**).

### 3.2. Spatial Changes in Reference Evapotranspiration and Trend Coefficient

Hilly regions located to the south of the Yangtze River are large and wide, featuring complex and diverse terrains as well as densely-distributed waters. As different landscape types result in differences in the spatial distribution of climatic variables, these differences will always influence the spatial distribution of reference evapotranspiration. As shown in Figure 5, the reference evapotranspiration in northwest and southeast in spring was relatively higher, and the reference evapotranspiration in Southwest and central east was lower, with the trend coefficient between 0.31 and 0.6. Stations with a positive trend coefficient accounted for 89.9%. In most areas, reference evapotranspiration showed a rising trend. Only a handful of areas in the east had revealed a downward trend. In summer, reference evapotranspiration decreased from the southwestern to the northeastern area, and the northwest is also lower; most trend coefficients fell between −0.29 and 0.0, and the stations with a positive trend coefficient accounted for 47.8%. The scope of the rising reference evapotranspiration trend was comparable to that of its downward trend, with the rising trend mainly occurring in the northeastern and southwestern areas. In autumn, reference evapotranspiration was generally higher in the east than that in the west, with the trend coefficient between 0.01 and 0.4. Stations with positive trend coefficient accounted for 69.6%, indicating an overall rising trend of the regional reference evapotranspiration in autumn. However, reference evapotranspiration in the southeast and northeast was relatively higher in winter, which was similar to the spatial distribution pattern of reference evapotranspiration in spring. Meanwhile, most trend coefficients fell between 0.01 and 0.3, and stations with a positive trend coefficient accounted for 85.5%. Although reference evapotranspiration in most areas revealed a rising trend, stations showing a downward trend were mainly distributed in the northwestern areas.

Areas with low annual average reference evapotranspiration coincided with that of the seasonal reference evapotranspiration, but large differences remained in high-value areas. Annual average reference evapotranspiration was between 1134 mm and 1387 mm, and reference evapotranspiration

in the eastern region was higher than that in the western region, which coincided with the spatial distribution of reference evapotranspiration in autumn. Areas with lower annual average reference evapotranspiration were mainly distributed in the western mountains with higher altitudes, mainly due to the lower temperatures, higher relative humidity, smaller surface water evaporation and favorable moisture conditions [16], the annual average reference evapotranspiration in the hilly regions in southern China was mainly between 0.01 and 0.3, and stations with positive trend coefficients accounted for 84.1%, showing an overall rising trend. As areas with larger trend coefficients were mainly located in the western, northern [36] and northeastern regions, a significant rising trend could be observed in these areas. On the whole, reference evapotranspiration to the east of the hilly regions in southern China was higher than that in the western regions, and the reference evapotranspiration in the western mountains was relatively lower. In most areas, reference evapotranspiration showed a rising trend, and a gradual downward trend was observed in a handful of high-altitude areas including Mount Qixian and Mount Wuyi.

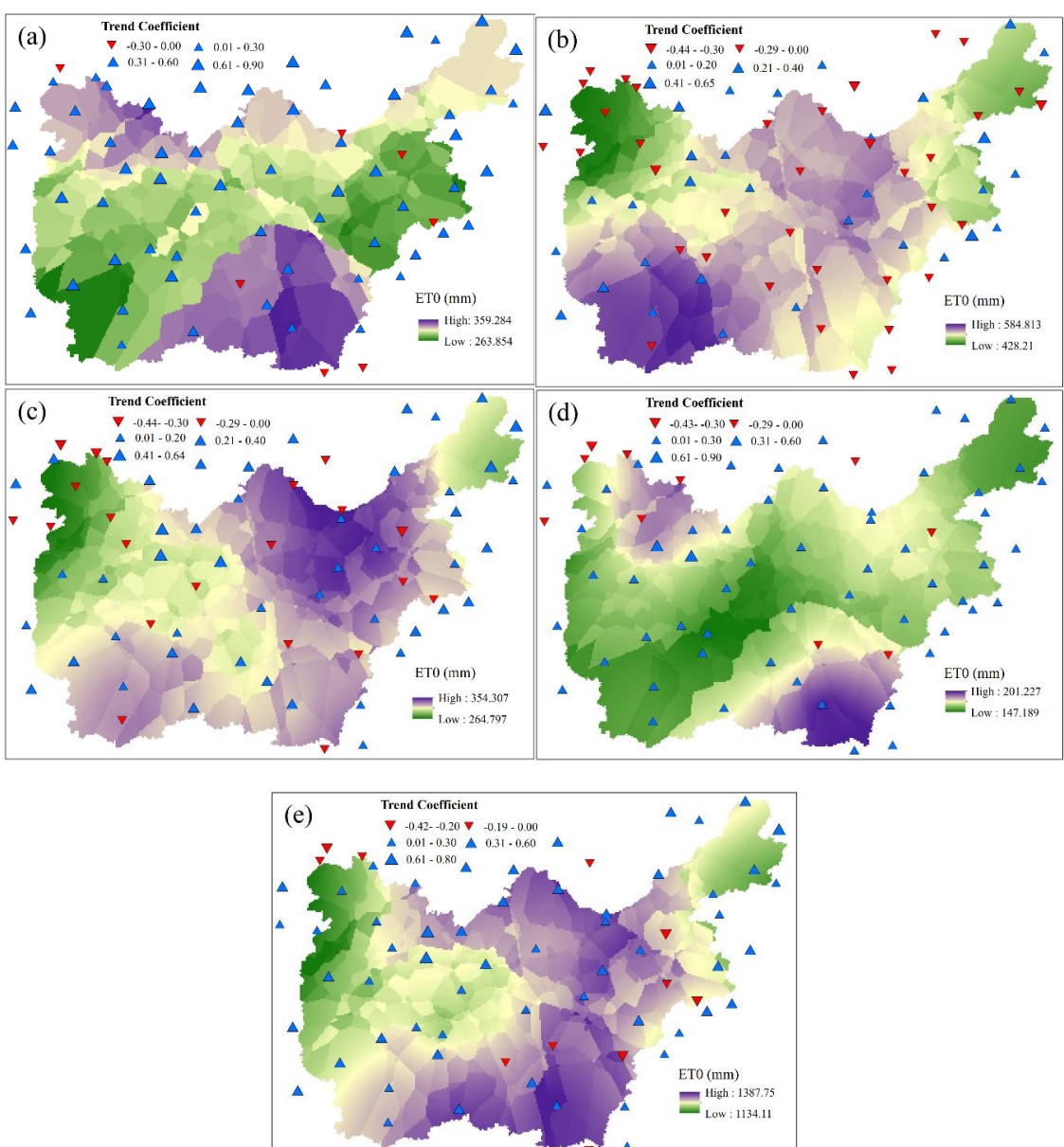

**Figure 5.** Spatial Distribution of Reference Evapotranspiration and Their Trend Coefficients of spring (**a**), summer (**b**), autumn (**c**), winter (**d**) and annual (**e**).

### 3.3. Analysis of Abrupt Changes in Reference Evapotranspiration

To conduct further discussion on changes in reference evapotranspiration, the MK (Mann–Kendall) Test Method was applied to analyze abrupt changes in reference evapotranspiration in the hilly regions in southern China over the last 63 years, since the MK test analysis method is relatively accurate for the mean mutation test, it was selected as the mutation analysis method, and the temporal sub-sequence was 5 years. For an introduction to this method, see the paper by Wei et al. [37]. The average reference evapotranspiration calculated from the multi-year average reference evapotranspiration of all sites in the region is used to represent the reference evapotranspiration status of the entire region, and the MK test of the reference evapotranspiration is performed. Figure 6 shows the MK Test of $ET_0$. UF referred to the sequential statistical curve (full line), and UB is the inverted sequential statistical curve (dotted line). If the value of UF or UB is greater than 0, the sequence indicates an upward trend, and less than 0 indicates a downward trend. The range in which they exceed the critical line is determined as the time zone in which the mutation occurs. If there is an intersection between the two curves of UF and UB, and the intersection is between the critical lines, then the moment corresponding to the intersection is the time at which the mutation begins [37]. Meanwhile, the significance level was set at 0.01, and the critical line was U = ±2.56. The UB and UF curves have a total of three intersections between the critical lines, which appeared in 1953, 1964, and 2008. Two important intersection points of UB and UF happened in 1964 and 2008 respectively, and the intersection points appeared in 1964 and 2008, also very close to the two time points (1966 and 2002) when the cumulative anomaly changes were significant (Figure 4b). The intersection point indicates that the starting point of mutation of reference evapotranspiration appeared in 1953, 1964 and 2008. This was mainly due to an abrupt change of meteorological variables that appeared at those times. The UB and UF are greater than 0 between 1953 and 1964 and after 2008, it indicates that the reference evapotranspiration has a sudden upward trend during these periods.

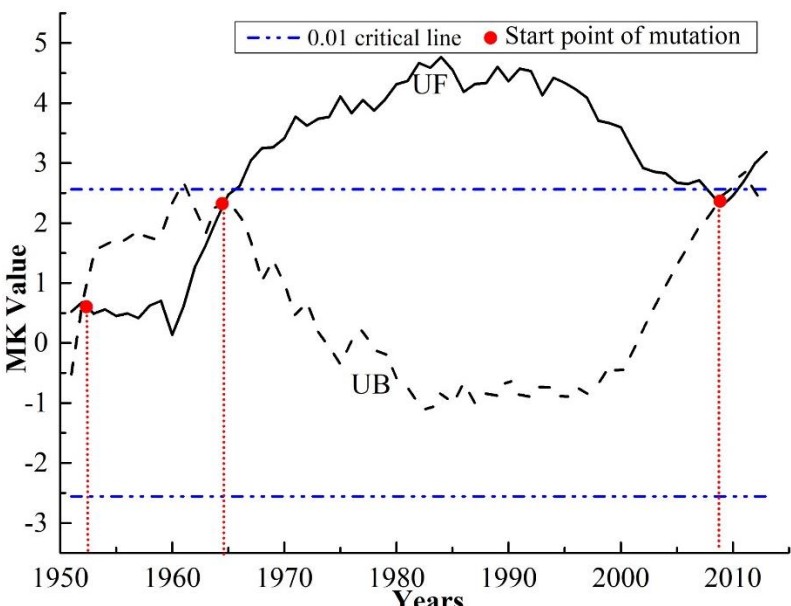

**Figure 6.** MK Test on Reference Evapotranspiration.

In order to further explore the temporal and spatial distribution of mutations in various meteorological stations, Figure 7 counts the number of mutations of reference evapotranspiration at each meteorological site and classifies the number of mutations by age. In total, 65 meteorological stations had shown abrupt changes in reference evapotranspiration, accounting for 94.2%. There was a significant fluctuation in reference evapotranspiration in most areas over the 63 years. The number of stations involved in such abrupt changes peaked in the 1950s and 1960s, indicating the significant sensitivity of

reference evapotranspiration in most stations in the 1950s and 1960s. This result is consistent with the result of Figure 6 that the mutation of the reference evapotranspiration in the entire region occurred in 1953 and 1964. Spatially, stations with the most abrupt changes were mainly distributed in the eastern region, especially in the southeastern and northern areas in the vicinity of Anhui Province. Only a handful of areas located in the northeastern and northwestern regions were free from abrupt changes. In contrast to Figure 5e, areas with the most abrupt changes in reference evapotranspiration coincided with that with large annual average reference evapotranspiration, showing that areas with larger reference evapotranspiration were also sensitive to changes in reference evapotranspiration.

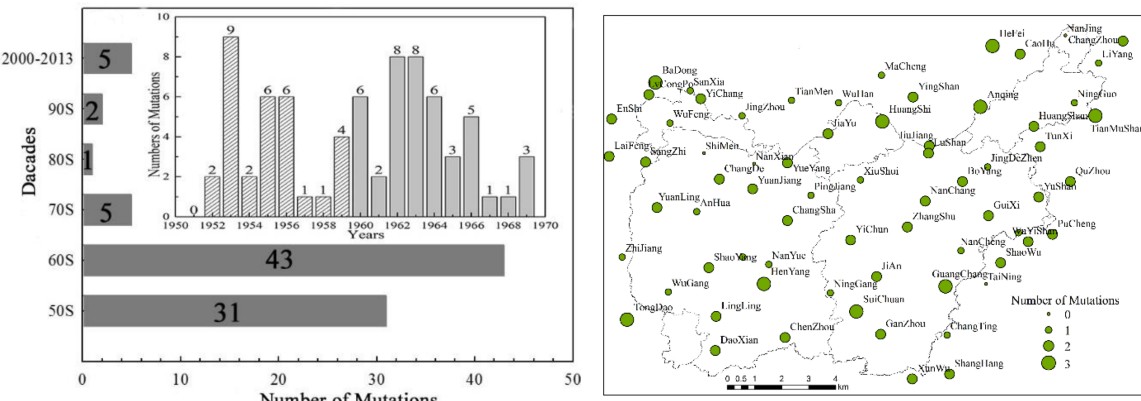

**Figure 7.** Spatial Distribution of Abrupt Changes in Reference Evapotranspiration at Various Stations.

*3.4. Periodic Analysis of Reference Evapotranspiration*

Similar to various climatic and hydrologic elements, reference evapotranspiration has corresponding periodic features as well. Thus, this paper has applied the continuous wavelet to analyze and discuss periodic changes in reference evapotranspiration in the hilly regions in southern China. As the degree of vibration revealed during the wavelet analysis indicates the stability of changes in reference evapotranspiration, where weak vibrations mean gradual changes while strong vibrations mean sharp changes. The periods of 1959–1966, 1972–1975 and 2001–2007 had vibrational periods of 1.5–4 years, 1.2–2 years and 2.5–3.7 years respectively during spring (Figure 8a). The periods of 1963–1973 had a vibrational period of 1.0–7.0 years during summer (Figure 8b). During the periods of 1966–1969 and 1994–2003, reference evapotranspiration in autumn had short periods of 1.2–2.0 years and 1.2–3.5 years periods respectively (Figure 8c). Vibrational periods of reference evapotranspiration in winter were 1.2–1.7 years, 1.2–2.0 years and 1.2–1.5 years for the periods of 1965–1968, 1997–2000 and 2005–2008 respectively (Figure 8d). According to the wavelet analysis on the annual average reference evapotranspiration, there had been a short 1.2–2.0 vibrational period from 1968 to 1974 (Figure 8e). In addition, the annual average reference evapotranspiration had experienced a relatively weak vibration in the end of the 1990s (since the 1970s), indicating that changes in reference evapotranspiration were stable during this period. This finding was similar to that of the annual average reference evapotranspiration shown in Figure 4b. In short, seasonal and annual average reference evapotranspiration had experienced 1.5–5.0 years of short-period changes. Based on the wavelet analysis on various meteorological variables, the periodic changes in reference evapotranspiration were in sync with that of relative humidity, proving that periodic changes in reference evapotranspiration were mainly caused by the short-period vibration of relative humidity.

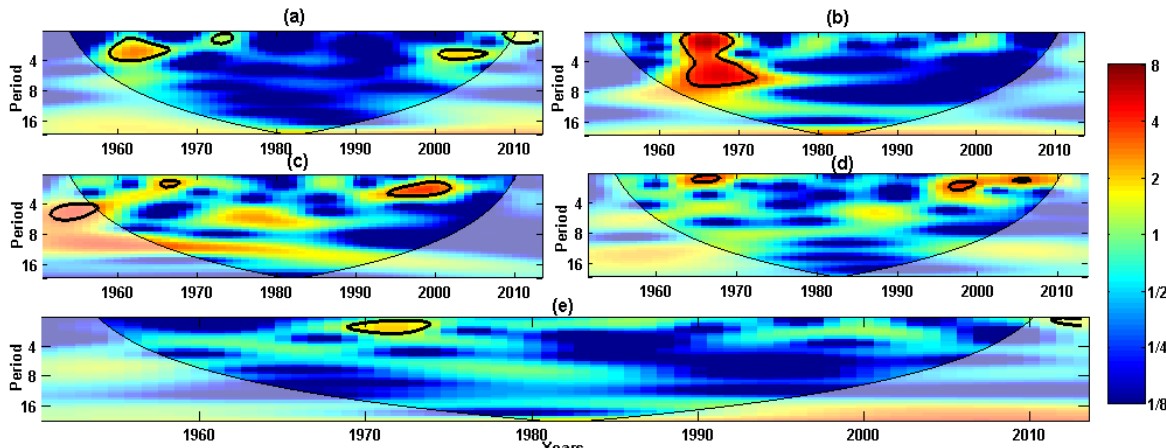

**Figure 8.** Wavelet analysis on spring (**a**), summer (**b**), autumn (**c**), winter (**d**) and annual (**e**) reference evapotranspiration.

*3.5. Analysis of Drivers of Reference Evapotranspiration*

3.5.1. Meteorological Variables

Pearson Correlation Analysis

Judging from the definitions of reference evapotranspiration, various meteorological variables, which influence these values include precipitation, percentage of sunshine, average wind speed, relative humidity and temperature [38]. Usually, the greater the percentage of sunshine, the longer the duration of actual sunshine; the higher the temperature, the faster the evapotranspiration of surface moisture. Similarly, an increase in wind speed would also accelerate evapotranspiration and may create significantly more moisture as well. Meanwhile, an increase in relative humidity would lead to a corresponding increase in surface water and moisture in the air, but reduce reference evapotranspiration [39]. As such, the Pearson correlation analysis was used to analyze the correlation between reference evapotranspiration and various meteorological elements [40] and r value was between [−1,1]. When $r > 0$, it indicates positive correlation, while $r < 0$ indicates negative correlation. The ranges of $0.3 > |r| \geq 0$, $0.5 > |r| \geq 0.3$, $0.8 > |r| \geq 0.5$ and $|r| \geq 0.8$ indicate weak correlation, medium correlation, strong correlation and extremely strong correlation respectively.

Based on Table 1, it is evident that precipitation has a negative correlation with reference evapotranspiration in the hilly regions in southern China on an annual and seasonal basis ($p < 0.01$). In addition, relative humidity was negatively correlated with reference evapotranspiration, and passed the 0.01 significance test. Although a weak positive correlation was observed on an annual basis and in autumn, it did not pass the significance test. In general, the wind speed was positively correlated with reference evapotranspiration, but, this correlation was not significant. Temperature was positively correlated with reference evapotranspiration. With the exception of an annual basis and in winter, sunshine percentage was positively correlated with reference evapotranspiration, and passed the 0.01 significance test.

**Table 1.** Partial Correlation between Reference Evapotranspiration and Various Meteorological Variables.

| Meteorological Variables | R Value | Time | Meteorological Variables | R Value |
|:---:|:---:|:---:|:---:|:---:|
| U | 0.034 | Annual | T | 0.593 ** |
|  | −0.61 ** | Spring |  | 0.652 ** |
|  | 0.017 | Summer |  | 0.457 ** |
|  | 0.029 | Autumn |  | 0.323 ** |
|  | −0.195 | Winter |  | 0.116 |
| RH | 0.076 | Annual | SP | 0.208 |
|  | −0.909 ** | Spring |  | 0.488 ** |
|  | −0.696 ** | Summer |  | 0.825 ** |
|  | 0.048 | Autumn |  | 0.392 ** |
|  | −0.853 ** | Winter |  | −0.17 |

** means it has achieved the significance (0.01) test level. RH is the relative humidity, U is the wind speed, T is the temperature and SP the sunshine percentage.

Sensitivity Coefficient and Contribution

Table 2 shows the relationship between the sensitivity coefficient of each meteorological variables, the amount of change over the years, and the contribution. Reference evapotranspiration has a higher sensitivity coefficient to the relative humidity and average temperature. Except to relative humidity, the sensitivity coefficient of reference evapotranspiration to temperature, wind speed and sunshine hours are positive. Many scholars have also proved that the sensitivity coefficient of reference evapotranspiration to relative humidity is usually negative in China [33,41–43]. Although the annual variation of average temperature and relative humidity is much smaller than the sunshine hours and wind speed, the reference evapotranspiration has higher sensitivity coefficient to the average temperature (positive effect) and relative humidity (negative effect), while the sensitivity coefficient to the other two meteorological variables (positive effect) are too small. As a result, the highest contribution is the average temperature, followed by the relative humidity. Many scholars have also shown that relative humidity has a great contribution to reference evapotranspiration in most areas of China, but the contribution of temperature to reference evapotranspiration is small in most parts of China as different locations, the contribution of wind speed to reference evapotranspiration is greater than temperature in Northeast China and Northern China [41,42]. The average temperature and relative humidity are positive contributions to the reference evapotranspiration. It is similar to the Anhui province of China, which is closer to the study area [33]. The sunshine hours and wind speeds are negative contributions to the reference evapotranspiration, and the negative contributions of sunshine hours to the reference evapotranspiration is much greater than the wind speed. The total contribution of the four main variables affecting the reference evapotranspiration to the change of reference evapotranspiration is 7.84%, while the average annual change of reference evapotranspiration in the past 63 years is 11.13%, indicating that the change of other variables may contribute positively (3.288%) to the reference evapotranspiration.

**Table 2.** The Sensitivity Coefficient and Relative Change of Meteorological Variables and Their Contribution to $ET_0$ Changes.

| Meteorological Variables | Sensitivity Coefficient | Relative Change (%) | Contribution Ratio (%) |
|:---:|:---:|:---:|:---:|
| T | 1.136 | 6.05 | 6.873 |
| n | 0.203 | −17.11 | −3.47 |
| U | 0.024 | −20.92 | −0.502 |
| RH | −1.098 | −4.5 | 4.941 |
| $ET_0$ | - | 11.13 | 7.842 |

Note: T means average temperature; RH means relative humidity; n means sunshine hours; U means wind speed.

Figure 9a shows the spatial ratio of the sensitivity coefficient of reference evapotranspiration to meteorological variables (average temperature, relative humidity, sunshine hours and wind speed) in the hilly area of Jiangnan. In space, the meteorological variables with strong sensitivity are temperature, relative humidity, sunshine hours and wind speed, but the sensitivity coefficient of $ET_0$ to wind speed is extremely low. Therefore, the spatial distribution of sensitivity coefficient of $ET_0$ to average temperature, relative humidity and sunshine hours is mainly considered. The sensitivity coefficient of $ET_0$ to the average temperature in the study area is positive (Figure 9b), shows that the reference evapotranspiration increases with the increase of the average temperature. Among them, the sensitivity coefficient of $ET_0$ to the average temperature in the north is generally larger than that in the south, and the sensitivity coefficient of $ET_0$ to the average temperature in the east is significantly larger than that in the west, indicating reference evapotranspiration is more sensitive to average temperature in the north and east. The sensitivity coefficient of $ET_0$ to relative humidity in the study area is negative (Figure 9c), indicating that the reference evapotranspiration decreases with the increase of relative humidity. The sensitivity coefficient' absolute value of $ET_0$ to the relative humidity in the northwestern region is generally smaller than other regions, indicating that the reference evapotranspiration in the northwestern has lower sensitivity with relative humidity (RH). The sensitivity coefficient of $ET_0$ to sunshine hours in the study area is positive (Figure 9d), the spatial distribution of the sensitivity coefficient of $ET_0$ to sunshine hours is different from the spatial distribution of average temperature and relative humidity, and the spatial distribution of sensitivity coefficient of $ET_0$ to sunshine hours is uniform, indicating that the sensitivity of reference evapotranspiration to sunshine hours is less affected by spatial location.

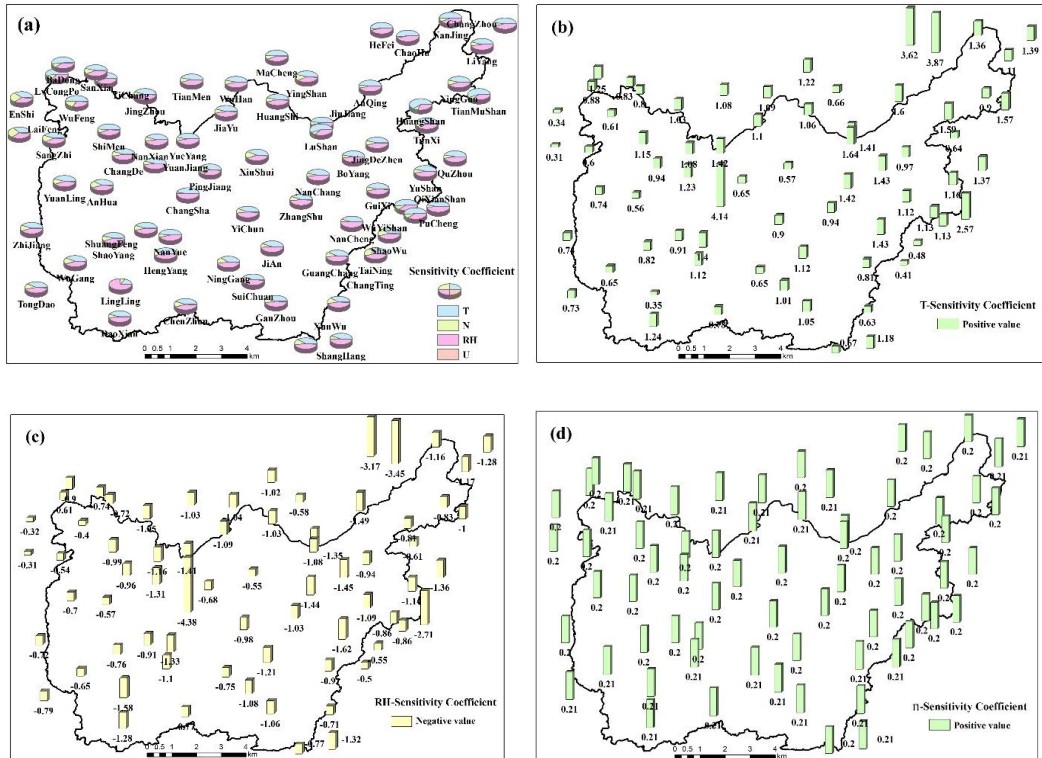

**Figure 9.** The Spatial Distribution of the Sensitivity Coefficient of Meteorological Variables (**a**), T (**b**), RH (**c**) and n (**d**) to Reference Evapotranspiration.

Figure 10a shows the spatial ratio of the contribution of meteorological variables (average temperature, relative humidity, sunshine hours and wind speed) in the hilly area of Jiangnan. Affected by the size of the sensitivity coefficient, the meteorological variables with a large contribution to the reference evapotranspiration are average temperature, relative humidity, sunshine hours, and wind

speed. Since the wind speed has a very small sensitivity coefficient to reference evapotranspiration, the contribution of the wind speed to the reference evapotranspiration is also small. Therefore, the spatial distribution of the average air temperature, relative humidity and sunshine hours to the reference evapotranspiration contribution is mainly analyzed. The contribution of the average temperature to the reference evapotranspiration is positively contributed by most sites, while the sensitivity coefficient of the average temperature is positive in Figure 10b. According to the Equation, the relative change of temperature in many sites is positive. The spatial distribution law of the contribution of average temperature is similar to the sensitivity coefficient of average temperature. The contribution of the north and east is greater than that of the south and the west, and the contribution of the southwest is the smallest. Relative humidity has a negative contribution to the reference evapotranspiration in most sites. Combined Figure 10c with Equation (10), the sensitivity coefficient of relative humidity is negative, indicating that the relative humidity of the study area has had a positive change over the years. The spatial distribution of the contribution of relative humidity was different from its sensitivity coefficient, and the contribution of relative humidity in the southeast is lower. The contribution of sunshine hours to reference evapotranspiration in space is mostly negative, while the sensitivity coefficient of sunshine hours in Figure 10d is positive, shows that the average change of sunshine hours is negative. The spatial distribution of the contribution of sunshine hours is far less uniform than the spatial distribution of its sensitivity coefficients, and the contribution of sunshine hours in the southwest is the smallest, mean that the average change of sunshine hours in the southwestern region also smallest.

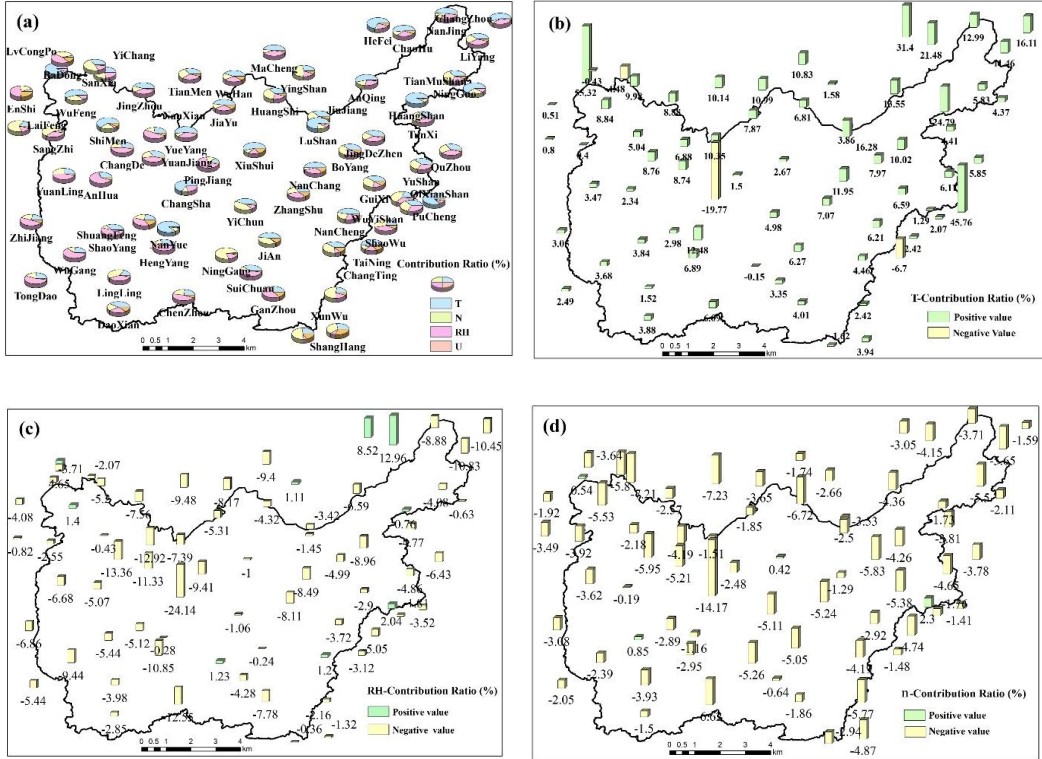

**Figure 10.** The Spatial Distribution of the Contribution of Meteorological Variables (**a**), T (**b**), RH (**c**) and n (**d**) to Reference Evapotranspiration.

### 3.5.2. Climatic Indicators

Reference evapotranspiration was affected by various variables. Driven by various meteorological variables, they also have a teleconnection with climatic indexes. From Table 3, a significant correlation existed between WP, SOI, TNA, and ENSO, and the reference evapotranspiration on an annual and seasonal basis.

**Table 3.** Correlation between Reference Evapotranspiration and Various Climatic Indexes.

| Climate Index | Spring | Summer | Autumn | Winter | Annual |
|---|---|---|---|---|---|
| WP | −0.343 ** | −0.016 | −0.005 | −0.097 | −0.122 |
| SOI | 0.186 | 0.183 | 0.221 | 0.306 * | 0.273 ** |
| TNA | 0.282 * | 0.112 | 0.338 * | 0.131 | 0.429 ** |
| ENSO | −0.243 | −0.032 | −0.242 | −0.37 ** | −0.205 |

** means it has achieved the significance (0.01) test level, and * the significance (0.05) test level. Others mean that it has not passed the significance test. WP: Western Pacific Index; SOI: Southern Oscillation Index; TNA: Tropical Northern Atlantic Index; ENSO: El Niño-Southern Oscillation.

Reference evapotranspiration was negatively correlated with WP on an annual and seasonal basis due to WP being closely related to the winter monsoon in the Northern Hemisphere. The stronger the WP in winter and spring, the lower the temperature, the larger the precipitation, the smaller the reference evapotranspiration [32,44]. Reference evapotranspiration and SOI were positively correlated on an annual and seasonal basis, and the significance test was passed on annual basis in winter, indicating that the stronger the SOI, the larger the reference evapotranspiration. Reference evapotranspiration was positively correlated with TNA on an annual and seasonal basis, and the significance test was passed on an annual basis, in spring and autumn, due to the positive correlation of TNA in summer with the South China Sea monsoon [44]. Reference evapotranspiration was negatively correlated with ENSO on an annual and seasonal basis, and the significance test was only passed in winter due to the significant increase in precipitation during most seasons in most parts of Southern China during El-Nino [45]; precipitation was negatively correlated with reference evapotranspiration. The correlation between reference evapotranspiration and ENSO was contrary to that between reference evapotranspiration and SOI indexes as SOI indexes were negative during El-Nino, and El-Nino3.4 Zone was negatively correlated with SOI indexes [45].

### 3.5.3. Other Drivers

Besides meteorological elements and climatic factors, the geographical distribution of surface water was also affected by the geographic location. By screening various geographic factors related to surface water and applying the multivariate regression analysis, the prediction models of annual and seasonal reference evapotranspiration in the hilly regions in southern China were created [46]. All five models had passed the significance test with confidence coefficient at $\alpha = 0.01$.

Analyses of the models showed that annual and seasonal reference evapotranspiration was negatively correlated with the altitude, indicating that a rise in altitude would cause a decline in reference evapotranspiration. As for latitude, a negative correlation could be observed in all seasons other than spring. This means that an increase in latitude would cause a decline in reference evapotranspiration as higher latitudes would lead to lower temperatures with lower evapotranspiration rates. With regards to longitude, a weak positive correlation was found, and an increase in longitude would cause an increase in reference evapotranspiration (Table 4). These laws are consistent with the contents shown in Figure 5. The reference evapotranspiration is relatively lower in areas with high altitudes; the reference evapotranspiration in the south is higher than in the north in the spring, summer, winter, and year; the seasonal and annual average reference evapotranspiration are higher in the east than in the west. Figure 11 shows the Variation Fitting for Annual and Seasonal Simulated Reference Evapotranspiration and Actual Reference Evapotranspiration with Latitude, Longitude and Altitude (69 stations were numbered from the lowest latitude, longitude and altitude to the highest as the horizontal axis). Most estimated values of annual and seasonal reference evapotranspiration fell within the relative error range (±20% of the actual measured values), proving that it is better to use regression distribution models to simulate changes in reference evapotranspiration with latitude, longitude and altitude. However, the annual fitting equation could achieve the best effect, while the effect of such a fitting equation in winter was slightly poor. Compared to latitude and longitude,

altitude would almost exert a synchronized influence on reference evapotranspiration both on a seasonal and an annual basis. On the contrary, reference evapotranspiration would decline as altitude rose. In particular, a sharp decline was observed from No.58 station (altitude above 350 m).

**Table 4.** Regression Distribution Models for Reference Evapotranspiration.

| Season | Regression Model | R Value |
|--------|------------------|---------|
| Annual | $1297.536 - 13.49x + 4.158y - 0.387z$ | 0.850 ** |
| Spring | $215.771 + 2.743x + 0.287y - 0.101z$ | 0.593 ** |
| Summer | $590.850 - 7.886x + 1.63y - 0.16z$ | 0.601 ** |
| Autumn | $82.742 - 4.708x + 3.404y - 0.096z$ | 0.687 ** |
| Winter | $163.404 - 2.722x + 0.795y - 0.032z$ | 0.408 ** |

** means it has achieved the significance (0.01) test level. X latitude, Y longitude, Z altitude.

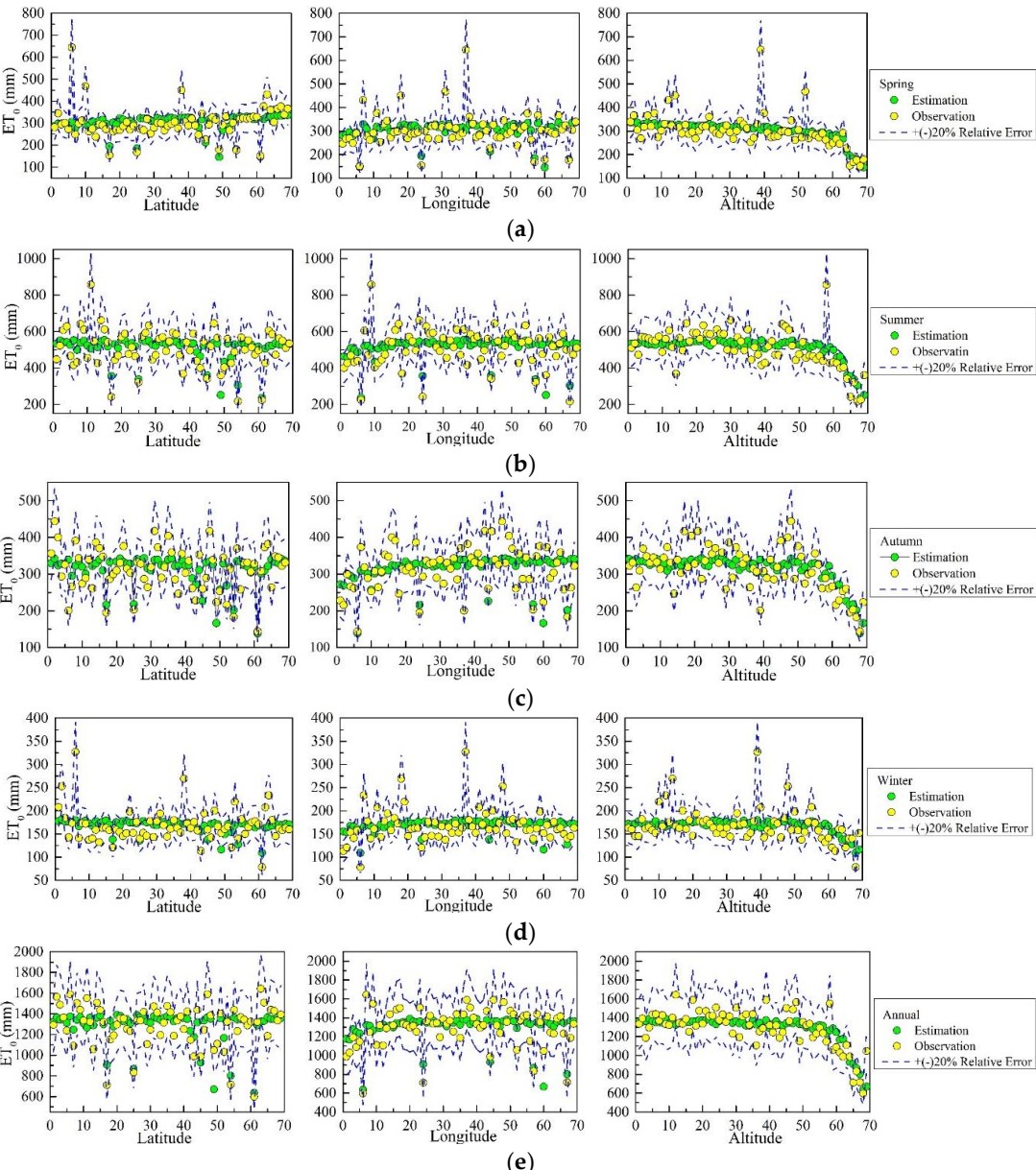

**Figure 11.** Variation fitting for reference evapotranspiration with latitude, longitude and altitude of spring (**a**), summer (**b**), autumn (**c**), winter (**d**) and annual (**e**).

## 4. Summary and Conclusions

Between 1951 and 2013, changes in the annual average reference evapotranspiration in hilly regions in southern China had featured a gradual fluctuating rise–gradual fluctuating decline–rapid rise pattern. However, an overall gradual rising trend could be observed; reference evapotranspiration reached its peak in summer and its lowest point in winter, and the rising trend was extremely significant in spring. Reference evapotranspiration to the east of the hilly regions in southern China was higher than that in the west, a rising trend could be observed in most areas and the reference evapotranspiration in a handful of high-altitude mountains was relatively lower with a gradual downward trend.

In 1953, 1966 and 2008, abrupt changes occurred in reference evapotranspiration. In fact, 94.2% of the meteorological stations were involved in such abrupt changes in reference evapotranspiration in the 1950s and 1960s; As the spatial distribution of stations involved in the abrupt changes in reference evapotranspiration was different, stations that were involved in the most abrupt changes in reference evapotranspiration were mainly distributed in the eastern region. Reference evapotranspiration was affected by periodic changes in relative humidity, and showed short-period vibration in most cases.

Various meteorological variables had played a driving role in reference evapotranspiration. Amongst these variables, relative humidity was negatively correlated with reference evapotranspiration on the whole, while temperature and sunshine percentage were positively correlated with reference evapotranspiration. The meteorological variables that have a great influence on the sensitivity of reference evapotranspiration are average temperature, relative humidity, sunshine hours, and wind speed. The sensitivity coefficient of reference evapotranspiration to temperature, sunshine hours and wind speed are positive, while the sensitivity coefficient to relative humidity is negative. It is roughly consistent with the correlation of reference evapotranspiration and various meteorological variables. The sensitivity coefficient of reference evapotranspiration to average temperature in the northern and eastern regions are larger than that in the south and west, and the sensitivity coefficient to relative humidity in the northwest is the smallest. However, the spatial distribution of the sensitivity coefficient of reference evapotranspiration to sunshine hours is relatively uniform, its less affected by spatial location. The spatial distribution of the contribution is similar to sensitivity coefficient, however, the spatial distribution of the contribution of sunshine hours is far less uniform than its sensitivity coefficient due to the difference of relative changes over the years. WP and ENSO are positively correlated with reference evapotranspiration, while SOI and TNA are negatively correlated with reference evapotranspiration. In addition, a negative correlation, weak positive correlation and negative correlation could be observed between the reference evapotranspiration and the latitude/longitude/altitude respectively.

**Author Contributions:** Conceptualization, Y.L. (Youcun Liu) and Y.L. (Yan Liu); methodology, Y.L. (Youcun Liu); software, Y.L. (Yan Liu); validation, D.L., Y.L. (Yongtao Li) and Q.D.; formal analysis, Y.L. (Yan Liu); investigation, X.B.; resources, Y.L. (Yongtao Li); data curation, Y.L. (Youcun Liu); writing—original draft preparation, Y.L. (Yan Liu); writing—review and editing, Y.L. (Youcun Liu); visualization, D.L.; supervision, D.L. and M.C.; project administration, Y.L. (Youcun Liu); funding acquisition, M.C.

**Funding:** This work is partially supported by the National Natural Science Foundation of China (41,861,002, 51,664,025), the Natural Science Foundation of Jiangxi Province (20181BAB203026); the Major Basic Research Development Program of Ganzhou City ([2017]179); Scientific Research Foundation for Qingjiang Scholars of Jiangxi University of Science and Technology (JXUSTQJBJ2017002).

**Acknowledgments:** We would like to express my gratitude to all those who have helped us during the writing of this thesis. We are also deeply indebted to all the tutors and teachers in Translation Studies for their direct and indirect help to us.

**Conflicts of Interest:** The authors declare no conflict of interest.

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
