# Peer review of "Characteristics and Drivers of Reference Evapotranspiration in Hilly Regions in Southern China"

_water, doi:10.3390/w11091914_

Round 1
Reviewer 1 Report
Thank you for your efforts in revising the manuscript.
Author Response
The author further revised the content of the article. Thank you for your careful review and suggestions. These reminders and suggestions give the author a lot of inspiration and give the author an opportunity to improve this article.
Reviewer 2 Report
I thought that resubmitted paper was a significant improvement over the original submission, and most of my previous issues have been adequately addressed. There are a few remaining issues where the author and I disagree, but I believe the author has articulated their position; and do not feel these are absolute requirements;
2.2.2 I cannot see how accumulative anomally helps describe trends.
Figure 5. It would be better if all figures had the same color scheme and legend, so that the colors and symbols meant the same thing on all figures.
3.5.3 These regression models supposedly predict ETo for any x,y,z in the image, therefore it would be more illuminating to compare the map of this regression for the whole region to the kriged output from Figure 5, so that the differences between the two could be seen where they are spatially.
Figure 11, not sure about it's usefulness compared to a single figure as above.
There were only a few issues in editing that I found in this review.
Everywhere) Please standardize references to figures and sub-figures, sometimes its a) Fig 4a. b) figure2a c) figure 2a, etc. My preference is Fig 4a, and less parens when possible.
128) radiation at the top of atmosphere.
218) ... variables from the ~70 stations in the hilly regions ....
311) .. Figure 4 shows MK test results ?
487) Strange font for smaller
3.5.1.2, should you use n and not N for sunshine hours? The current name differs from the Rn calculation.
Author Response

(The authors gave the same response as above.)

Reviewer 3 Report
The manuscript by Liu et al., (2019) presents a study on investigating drivers and features of reference ET in southern China. Due to the importance of ET for different research fields, it has been widely studied at different spatial and temporal scale over different areas in the world. The good parts of the current study are that it has made a comprehensive analysis on ET trend and distribution and related drivers. The current study needs to be better presented to describe its contribution to the research community. My detailed comments are listed below.
The introduction needs to be improved to better state the new parts of the current study and add state-of-the-art references related to your study.
Careful analysis of ET is presented in the manuscript. The findings of the study suggest the dominant drivers on ET are temperature, wind speed, relative humidity, which are not surprising. And they have been known and found by many studies. I would suggest the authors compare their results and previous studies. The differences and similarities should be compared and discussed.
Regarding ET and reference ET, the authors need to add a few sentences to explain them. In particular, reference ET refers to the conditions where no water limit exits. It means the analysis here suggests only the conditions that are determined by meteorological variables rather than surface conditions. For many fields, ET rather than reference ET is more important. The authors should add discussions on the perspective.
Author Response

(The authors gave the same response as above.)

Reviewer 4 Report
refer to attached file

Author Response
The author further revised the content of the article.
Thank you for your careful review and suggestions. These reminders and suggestions give the author a lot of inspiration and give the author an opportunity to improve this article.
Please see the attachment for specific reply.

Round 2
Reviewer 3 Report
Thanks for your making efforts to revise the manuscript according to my comments. I have no further comments and recommend you carefully check the language again before considering for publication.
Reviewer 4 Report
The authors modified the paper based on the comments for the 1st round review and I believe that it has satisfactory improved.
This manuscript is a resubmission of an earlier submission. The following is a list of the peer review reports and author responses from that submission.
Round 1
Reviewer 1 Report
Overall
This paper is primarily an atlas of various ETo and Water Balance parameters over fairly large temporal and spatial region in China, and can be used to 1) infer some general trends of ET and water balance 2) Provide some introductory insight to other users on the viability of certain methodologies. For the most part, the methodologies are fairly simple, and can be easily replicated.
The paper did not give me any particular insight best practices for application to new areas. Many of the figures were hard to read; a lack of a strong research methods section; and no conclusions on the process itself limited it's more general applicability.
The paper had no major issues in grammer, syntax,etc.
2. Data and Methods
This section is weak. No description of interpolation, no description of Mann-Kendall testing, no description of climatic indicators. No description of Wavelet analysis. These are never adequately introduced in the paper, and limit it's usefullness
2.2.2 At least describe the difference between ET and ETo and how this affects real water balance. K is basically an Ag indicator.
3. Results
It's not clear if the overall (regionally) figures are the sum of all the stations or a sum over the region. SInce the interpolation is Inverse distance, that would be a simple weight on each station . Would like to know the range of those weights.
FIgure 2. (Can't read legends)
Line 182: Confusing
LIne 185: Redundant
Figure 3. (Bad Legend), I can't come up with an example in my head where cumulative anomaly adds to these figures. (or is that useful)
Line 208: And all places. If you add the title of the figure in addition to the figure number, the please italics if that's allowed.
Lines 210-226: I will comment that these lengthy descriptions of the figures do not help me in particular, and seem to hide major conclusions. I note that that could just be me however. (This applies to most figures)
Figure 4. I think the authors could choose a single color legend that would be applicable to all figures, and give us insight into the trends between seasons. This is true for both the area and the stations. The figure also shows the problems with using IDV for interpolation over a hilly region. Since this is a semi-conclusion later, this should be discussed in results. Also, look at these anomalies (eg NanYue vs HenYang stations) That's a good indicator of the limits of this interpolation.
Figure 5. Same comment on legends
3.3 MK test never discussed. Lot's of stuff like significance level, critical line, etc with no background. As a result Figure 7 not useful
LIne 293: Don't know what (a) after years mean.
Figure 7. No legend
Figure 8. I'm highly suspicious about this sudden shift in changes afger the 60s. Could this be a factor from something else, like bad records?
3.4 The wavelet analysis seemed like an interesting test, but that lack of description on the method, plus the lack description of the results didn't provide mush insight on what those figures were showing.
Lines: 359-361. No supporting documentation for that claim.
Lines: 377-379. No supporting documentation of that claim.
3.5.1 I have some concerns about the way this whole section is presented. We are *calculating* ETo from these parameters, we don't need to estimate correlations, we can *calculate* correlations. What you are trying to say is that a certain station gives you an ensemble of parameter sets (each day) , and you are looking at what parameters from that ensemble best indicate changes for that station an season. You could do this other ways (eg calculate the PM from the mean values, and then vary each parameter 2 sigs on that PM calculation) . I'm not a statistication, so I can't say f pearson correlation is appropirate.
Line 420: Introduce the term "Contribution Ratio" without definition. I don't understand, eg RH is certainly not the most important term in ETo calculation, are you saying the variations that you saw in the station data lead to the largest *changes* in ETo?
Figure 11,12. Why is the background changing.
3.5.2. I just skipped this section, since there were no definitions of WP, SOI, TNA, ENSO where what we'd expect to see. Need a better methods section,
3.5.3. Lat-Long is not the best coordinate system choice.
It would be useful to compare for one season, the spatial map of the regression, vs. the IDV version you used in the text. It could inform us on the problems with the interpolation methods. Alternatively, you could replicate FIgure 2 using this regression for one season.
Figure 18. Can't read. You also seem to randomly flip the colors between observation and estimation. I'd rather see one map as described above to these graphsn
Figure 19. Why show if insignificant?
Reviewer 2 Report
Review: "Characteristics and Drivers of Surface Reference Evapotranspiration and Water Surplus/Shortage in Hilly Regions in Southern China"
The authors present a study of ETo, its trends, dynamics, and drivers for a South Eastern Chinese region with considerable topography using meteorological station data. While ET is important for agriculture and ecosystems, I am not sure how much this study will help with the stated goal of setting a baseline for water resource management.
In my opinion considerable work is needed before the manuscript should be considered for publication.
While the paper does describe trends and presents the impacts of some drivers, these sections should be expanded upon, to better elucidate environmental processes. The introduction lacks a clear description of the underlying processes and does not include many relevant general references. Methods are not sufficiently elaborated to allow reproduction of results or easy interpretation.
It is often not clear to me, what general scientific questions are being addressed, since the paper contains a large collection of different analysis, but very little interpretation and discussion of results.
Moderate language revision and improvement of figures (including captions is needed).
General/ Major Comments
The abstract should be shortened and refocused on the main results, rather than just stating which correlations are being calculated. The focus should be on results rather than methods and data used.
Introduction:
The introduction should be rewritten. At the moment the introduction mentions that ET is important, that ET has been studies globally, but that Jiangnan has not been studied, because it is in a humid climate. I don't think that this is sufficient.
The introduction should outline a clear motivation, why it is worth studying and should introduce the relevant processes and factors that affect evapotranspiration and the water balance. The introduction hardly gives any specific references for the problem at hand and relies on few application studies for other regions in China, which may or may not be accessible for an international audience.
Pokely (2009) is not a scientific publication but a report on Roderick
et al. There are many primary sources for ET trends in the literature. I
suggest that the authors use google scholar to retrieve primary
literature that has studies trends in ET and potential ET. A starting
point may be other papers that have cited Roderick & Farquhar
(2002).
Methods:
The method section is not sufficient to understand how data was collected and processes. For example, break points in the trends of the wavelet analysis are not mentioned here, but introduced in the results section (and not sufficiently there to potentially reproduce results)
A note on "reference evapotranspiration" > Reference evapotranspiration or ETo refers to evapotranspiration from a well watered crop with specified roughness and specified surface resistance, as such it is akin to potential evapotranspiration and NOT actual evapotranspiration. This is important for real world applications, since potential evapotranspiration can be much larger than actual transpiration during times with water stress. This should be made clear in text and will also affect interpretation of results.
Rn and G are rarely measured, but required in eq1. How are they estimated?
Are the correlations calculated based on monthly or daily data?
Results:
The authors calculate seasonal trends of ETo for various meteroroligical stations. It would be good to also show the annual cycle, rather than just trends for individual seasons.
May main takeaway from this analysis is that spring ETo has been increasing, which in then turn impacts the water balance. My question would be, are these trends statistically significant? If so, the results of statistical tests should be presented.
The authors also present a cumulative anomly, but it is not clear to me with respect to what reference this anomaly is calculated. Is it with respect to the trends. It is not clear to me what the cumulative anomaly represents, especially with respect to the surface water balance. For the majority of years there is a water surplus, however the cumulative anomaly at the end of the period is zero. What is the significance of this?
Spatial interpolation of trends: This seems to be done with a simple distance based interpolation. I don't think that this is a sensible approach, given that the authors stress the impact of topography, which greatly modifies weather variables. I suggest that the authors either remove this or find a more sensible interpolation that takes into account terrain. Also, Figure 4 is generally hard to understand. There are a few outlier stations, which have very different ETo than the surrounding stations and then there is also the trend being indicated. Units are missing for both ETo and for the trend. The color-scale of the figure 4 and 5 should be made the same accoss subplots.
Section 3.3. Change point analysis
The methodology behind this section should be explained in the Methods section. At the moment some information is given, but it is hard to understand what exactly was done. Also, I don't fully understand the value of this analysis. What is the main result and why is this important in the contect of ETo and surface water balance. There is not enough information given in Figure 8 to easily understand what it means. I assume that it corresponds the the number of potential change points per decade (Mutations is almost certainly the wrong English word). I am also wondering whether some of the changepoints are related to data quality rather than to underlying physical processes there may have been significant changes in instrumentation and environment of the meteorological stations during the last 70 years.
Section 3.4
-Vibration is the wrong term. Timescale?
- I am asking myself the following from section 3.3. and 3.4: How does this help me understand the system, except that there is some periodicity or temporal variation, which is intrinsic to any natural system?
Section 3.5.
I think that this is the most interesting section of the paper, since it starts answering the question, what drives the observed changes. I would shorten/ remove sections 3.3/3.4 and expand this section. Also, the methodology should be updated accordingly to show what exactly was done.
Table 1 should be clearly deliniated between variables.
Section 3.5.2 Climatic indicators
Table 2. What do the stars mean?
Also, the authors calculate the correlation between ETo and WB and several climatic indices. These indices are never introduced nor is it made clear, why this is relevant. A deeper analysis, which is in my opinion needed would include an investigation of how these indices impact forcing variables of ETo and WB. This is partially done in the text, but should be expanded upon, especially with respect to the individual stations and the relative strenght of the impact.
3.5.3 Other drivers.
Specifically, this section applies a multi-linear regression using lat, lon, and height msl to model ETo and WB. While this may work well with the stations being considered, to be useful it would be important to know whether this would hold true if new stations were being added to the station. Similarly, is the still true under a climatic trend, or do these relationships change with time. One way to address this would be to split the data randomly into a training (for calculating the regression) and a test data-set (which is not used in the regression ) and then to evaluate the performance of the model on the test set.
Figures 13 and 14 are too small to be easily read.
Specific/ Minor Comments (not exhaustive)
P1L29: " as shown in 1956, 1962 and 2007" > I don't know what is being shown
P1L30: "evapotranspiration and water surplus/shortage in the east were both higher" > What is higher shortage or surplus. It is counterintutive that both are higher.
P1L61: "water surpluse/shortage" > this is very generic, consider using "surface water balance", when both are meant and either surplace or shortage, if talking about specific results.
P1L68 "Quite a few studies have been conducted on reference evapotranspiration and water surplus/shortage at home and abroad " ? where is home and abroad. The journal is targeted at and international audience.
Technical Comments (not exhaustive)
P1L18 > "This paper has adopted related ..." > This work uses data collected from 69 meteorological stations ... to <do what>?
Reference style should be homogenized (e.g. Zeng instead of ZENG, no firstnames in withintext citations, et al. XXXX instead of et al,XXXX; etc) . Also MDPI uses intext numbered citations.
P1L87: "Jiangnan hilly area" > is this the appropriate translation?, Consider Jiangnan Region, which sounds better to English native speakers.
L109: "meteorological factors" > meteorological variables
Figure legends and axis labels can be very hard to read due to size and figure quality.
WD: It is probably better to abbreviate wind speed as WS or U since WD is often used for Wind Direction.
Reviewer 3 Report
The study analyzed the characteristics and drivers of surface water changes in hilly regions located in southern China from the perspective of reference evapotranspiration and water surplus/shortage. The manuscript is overall well written, and it is easy to follow. I found the paper interesting and of interest to Water Journal. However, the paper is too long and I believe some of the results can be provided in an appendix or a supplementary file. The authors provide a short literature review of the previous studies, and thus the introduction section needs to be modified accordingly. Most of the paper figures require some modifications before publication. The authors may apply moderate revisions and make the study a strong addition to Water Journal. Please find more details in the second comment below.
Moderate issues:
Lines 18-52: Long Abstract!
Line 56: Section 1: Too short Introduction! Introduction section needs to be extended and modified thoroughly.
Line 172: Figure 2: the legends are not readable. Please make the font size bigger and suitable for the reader.
Line 200: Figure 3: Please provide plots with bigger font size to make it suitable for the reader.
Lines 227-228, 250-251, and 276: Figure 4, 5, and 6: Same issues of font size! Please modify.
Also, I think it is better and easier to follow the plots by ordering them as a, b, c, and d in figures 4 and 5; a and b in figure 6. Please consider modifying this in the text as needed. (Optional modification)
Line 293:” 63a years, and the temporal sub-sequence was 5a”: What does letter “a” next to number of years mean?
Line 307: Figure 7 caption: Please mention what UB and UF are referring to in figure 7 caption and not only in text.
Lines 431- 432 and 445-446: Figure 11and 12: Font size!
Line 454: Define WP, SOI, TNA and ENSO
Figure 13 and 14: Font size and please modify the last plot in the first row in both figures.